# Tracing Inflammation in Ischemic Stroke: Biomarkers and Clinical Insight

**DOI:** 10.3390/ijms26199801

**Published:** 2025-10-08

**Authors:** Gaetano Pacinella, Mariarita Margherita Bona, Federica Todaro, Anna Maria Ciaccio, Mario Daidone, Antonino Tuttolomondo

**Affiliations:** 1Internal Medicine and Stroke Care Ward, Policlinico “P. Giaccone”, 90127 Palermo, Italy; mariaritamargherita.bona@gmail.com (M.M.B.); federicatodaro21@gmail.com (F.T.); amciaccio21@gmail.com (A.M.C.); mario.daidone@unipa.it (M.D.); bruno.tuttolomondo@unipa.it (A.T.); 2Department of Health Promotion, Mother and Child Care, Internal Medicine and Medical Specialties (PROMISE), University of Palermo, 90127 Palermo, Italy

**Keywords:** ischemic stroke, neuroinflammation, immunoinflammatory profile, inflammatory markers

## Abstract

Ischemic stroke is now widely recognized as a disease with a strong inflammatory profile. Cerebral vascular damage is both preceded and followed by a chain of molecular events involving immune cells and inflammatory markers, irrespective of the etiology of the ischemic injury. Over time, an increasingly comprehensive understanding of these markers has led to a better insight into the mechanisms behind the vascular event and recovery following ischemic stroke. However, to date, there are still no available circulating or tissue biomarkers for early diagnosis or prognostic stratification, making ischemic stroke diagnosis contingent on clinical and instrumental investigations. However, neurological and internal medicine research is progressing in identifying markers that could potentially take on this role. This manuscript, therefore, aims to review the most recent and innovative results of medical advances, summarising the current state of the art and future perspectives. If ischaemic stroke is an inflammatory disease, it is also true that it is not just a singular condition, but a group of entities with their own neuroinflammatory features. Thus, given that, in ischemic cerebral vascular damage, “time is brain,” tracking increasingly accurate markers in the diagnosis of ischemic stroke is a valuable tool that will potentially enable earlier recognition of this disease and, hopefully, make it less disabling and more widely treated.

## 1. Introduction

A stroke is a neurological deficit caused by an acute focal injury to the central nervous system (CNS) resulting from a vascular cause, including cerebral infarction, intracerebral haemorrhage (ICH), and subarachnoid haemorrhage (SAH). Advances in neuropathology and neuroimaging have improved the understanding of ischemia, infarction, and hemorrhage in the CNS. The pathophysiological mechanisms that cause a stroke can be of two types: ischaemic or haemorrhagic. Ischemic stroke results from the occlusion of an artery that delivers blood to the CNS, leading to a fast decline in oxygen and glucose delivery to the affected brain region. This event triggers a well-characterised sequence of patho-biological events—collectively identified as the “ischemic cascade”—involving bioenergetic failure, oxidative stress, disruption of ionic homeostasis, and, finally, neuronal cell death through necrotic and apoptotic pathways.

In contrast, hemorrhagic stroke arises from the rupture of intracranial vessels, manifesting as either intracerebral hemorrhage or subarachnoid hemorrhage. The ensuing leakage of blood not only exerts mass effect and increases intracranial pressure, but also causes a cascade of secondary injuries mediated by the toxic effects of blood components and the activation of innate immune responses [1]. Ischemic stroke is the clinical and nosological entity this manuscript focuses on: it constitutes a leading cause of mortality and long-term disability globally, representing a challenge for public health systems and placing a considerable burden on patients, caregivers, and healthcare infrastructures. The latest Global Burden of Diseases, Injuries, and Risk Factors Study (GBD) showed that in 2021, stroke was the third leading cause of death (GBD level 3), after ischemic cardiac disease and COVID-19 (7.3 million deaths; 10.7% of all deaths) and the second leading cause of death after ischemic cardiac disease globally. In addition, according to the same report, stroke is the fourth most common cause of disability-adjusted life year (160.5 million—5.6% of all DALYs). The prevalence of ischaemic stroke is 65.3%, thus categorizing it as the most prevalent form of cerebrovascular accident [2].

Over the years, clinicians and researchers have suggested various classification systems to categorize its etiology, focusing on pathogenesis or clinical presentation; the most used, for its ease and good interobserver agreement, is the Trial of ORG 10172 in Acute Stroke Treatment (TOAST) classification: it was introduced to standardize the stratification of ischemic stroke subtypes and to improve the understanding of pathogenic mechanisms, prognosis, and therapeutic responsiveness. The TOAST system categorizes ischemic strokes into five major subtypes, each defined by distinct pathogenic mechanisms (Table 1):-**Large-Artery Atherosclerosis (LAAS)** is a subtype that includes strokes due to atherothrombotic occlusion or stenosis of large extracranial or intracranial arteries, most commonly the internal carotid, vertebral, or middle cerebral arteries; the underlying pathophysiology involves plaque rupture, in situ thrombosis, or artery-to-artery embolism originating from an atherosclerotic lesion; diagnostic criteria require ≥ 50% stenosis of the involved cerebral artery, as evidenced by vascular imaging, with a compatible “infarct pattern” on neuroimaging. The lesions frequently exceed 1.5 centimetres in diameter; this subtype is associated with high recurrence risk and is closely linked to modifiable vascular risk factors, such as hypertension, dyslipidemia, diabetes mellitus and smoking.-**Cardioembolic infarct (CEI)** is a lesion caused by embolic occlusion of cerebral vessels arising from a cardiac source, often associated with high-risk cardioembolic conditions. These include atrial fibrillation, recent myocardial infarction, left ventricular thrombus, prosthetic heart valves, rheumatic mitral stenosis, infective endocarditis, cardiac tumours (such as atrial myxomas), and patent foramen ovale (PFO) in patients with deep venous thrombosis (DVT) (paradoxical embolism). The embolus typically results in a sudden onset of neurological deficit, accompanied by a large cortical infarct on neuroimaging, often without preceding transient ischemic symptoms. These infarcts are typically cortical, multifocal, or bilateral, and may involve multiple vascular territories, reflecting the embolic nature of the damage. Hemorrhagic transformation is more common in CEI than in other ischemic subtypes, which is attributable to both the size of the infarction and reperfusion injury following embolus migration or fragmentation. Cardioembolic stroke is associated with higher early mortality and recurrence rates. Diagnosis is contingent upon the identification of a plausible cardiac embolic source and the exclusion of significant large-artery disease or small-vessel pathology.-**Small-Vessel Occlusion (SVO), or Lacunar Stroke,** refers to infarctions resulting from occlusion of a small penetrating artery that supplies deep brain structures such as the basal ganglia, thalamus, internal capsule, or brain stem pons. The underlying mechanism is usually lipohyalinosis or microatheroma formation in the context of chronic hypertension or diabetes. Clinically, lacunar strokes are associated with classic lacunar syndromes (e.g., pure motor hemiparesis, pure sensory stroke) and typically present with small (<1.5 cm), deep infarcts on neuroimaging. There is no evidence of large-artery stenosis or a cardioembolic source.-**Stroke of Other Determined Etiology (ODE)** includes less common causes of ischemic stroke with clear and well-documented pathophysiological mechanisms, such as non-atherosclerotic vasculopathies (e.g., arterial dissection, vasculitis), hematological disorders (e.g., antiphospholipid antibody syndrome, sickle cell disease), and certain genetic or metabolic conditions. The identification of a rare but definitive cause necessitates a comprehensive diagnostic workup beyond routine stroke evaluation.-**Stroke of Undetermined Etiology (UDE)** is the category used when a definitive cause cannot be discovered, either due to an absence of comprehensive diagnostic investigations or because there are multiple potential causative factors. It is also pertinent to include cases where the stroke is confirmed by imaging, but the source remains unknown despite thorough diagnostic evaluation [3,4].

Although the TOAST system is most widely used in clinical practice, another classification approach is the ASCOD (A: atherosclerosis; S: small-vessel disease; C: cardiac pathology; O: other causes; D: dissection). This classification system has been developed to predict the likelihood of a causal relationship between diseases in the context of ischemic stroke; it is designed to comprehensively characterise the range of underlying diseases present in individual patients [5]. Neuroinflammation has been increasingly considered as a critical modulator of stroke evolution: the pathophysiology of ischaemic stroke involves several mechanisms that are common to all etiologies. Neuroinflammation, in fact, is not related to the etiology of the ischemic damage itself, but rather to the way in which the vascular accident affects the brain. Following an ischemic stroke, an inflammatory response arises in the CNS, characterised by the activation of microglia, the recruitment of peripheral immune cells, including neutrophils, monocytes, and lymphocytes, and the release of pro-inflammatory cytokines, such as IL-1β, IL-6, and TNF-α, as well as chemokines and danger-associated molecular patterns (DAMPs).

These mediators contribute to the breakdown of the blood–brain barrier, endothelial dysfunction, and the exacerbation of neuronal injury. Notably, inflammation in stroke pathophysiology exhibits a dualistic nature: while an excessive or dysregulated inflammatory response is detrimental and promotes neuronal injury, controlled and temporally orchestrated immune activity is essential for phagocytic clearance of cellular debris, the promotion of angiogenesis, and the facilitation of neurorepair mechanisms. This dichotomy highlights the importance of accurately defining the characteristics of neuroinflammation to identify optimal therapeutic windows in which to take action to modulate it beneficially: recent research has shown that neuroinflammatory pathways are crucial in the acute damage step and contribute to complications in the subacute and chronic phases of ischaemic stroke [6,7,8].

In this review, we present a comprehensive and critical examination of the current literature evidence on the inflammatory response of ischemic stroke, focusing on molecular mechanisms and biomarkers of neuroinflammation, their importance in prognostic stratification, and their implications for novel therapeutic targets. We further explore emerging immunomodulatory strategies aimed at limiting secondary brain injury and promoting functional recovery, thereby highlighting the translational potential of targeting inflammation in the context of cerebrovascular disease.

The scientific literature included in this review was selected through a comprehensive search of major scientific databases, namely PubMed and Google Scholar. The search focused on studies published in English and used a combination of specific keywords and Medical Subject Headings (MeSH) terms such as “neuroinflammation,” “ischemic stroke,” “immunoinflammatory profile,” “inflammatory markers,” and “immune response,”. Boolean operators (AND, OR) were used to combine search terms and narrow or expand the scope as needed.

To ensure the inclusion of relevant and high-quality studies, both recent publications (from the last 10 years) and older relevant works were considered. Additional references were identified through manual screening of bibliographies from key articles (backward citation tracking) and by exploring related articles suggested by the databases.

Only peer-reviewed articles were included, with priority given to original research papers, systematic reviews, and meta-analyses published in reputable journals. The collected literature was reviewed and selected based on its relevance to the pathophysiological mechanisms, immune responses, and clinical implications of neuroinflammation in ischemic stroke.

## 2. Molecular Mechanisms of Inflammation in Ischemic Stroke

Among the cells involved in the molecular mechanisms triggered by ischemic stroke, microglial cells play a decisive role. Neuroinflammation is primarily influenced by these cells, specifically resident macrophages in the central nervous system, which serve as a bridge between the central nervous system and the immune system [9].

Microglial cells are activated by damage signals released by impaired neurons (DAMPs); their activation results in pleiotropic effects: they remove cellular debris through phagocytosis, and produce molecular mediators (signalling molecules, cytokines, neuromediators, and extracellular matrix proteins) to regulate neuronal and synaptic activity and their functional plasticity [10].

Changes in microglia morphology or expression of cell surface antigens can cause rapid modifications in their phenotype in response to perturbations of CNS homeostasis. These cells, in fact, can get two main activation phenotypes: the M1 phenotype, which produces pro-inflammatory cytokines, such as IL-1β, IL-6, TNF-α, and neurotoxic mediators like reactive oxygen species (ROS) which can exacerbate neuronal damage, also leading to blood–brain barrier disruption and neuronal death; and the M2 phenotype, which stimulates anti-inflammatory responses, produces cytokines like transforming growth factor-β (TGF-β) and interleukin-10 (IL-10), which can help to inhibit neuronal damage and stimulate the repair of damaged tissues [11,12,13,14,15].

Studies have shown that the phenotypic polarisation of microglial cells depends on modulating the expression of the transcription factors interferon regulatory factor 4 (IRF4) and interferon regulatory factor 5 (IRF5). Specifically, downregulation of IRF5 leads to increased expression of IRF4, which promotes the M2 phenotype and improves stroke outcomes. Conversely, downregulation of IRF4 increases IRF5 expression, promoting the M1 phenotype and worsening stroke outcome. The balance between these two transcription factors is crucial in determining the response of microglia during and after an ischemic stroke, which significantly impacts the severity of brain damage and subsequent recovery [16] (Figure 1).

Among the most relevant mediators in the neuroinflammatory cascade are the chemokines, a set of proteins that enable immune cells to migrate from the bloodstream to the site of inflammation. These proteins and their receptors are crucial in the inflammatory response following ischemic stroke. The production of these molecules by microglial cells leads to several effects, including cell recruitment, alteration of blood–brain barrier (BBB) permeability, stimulation of neurogenesis, and promotion of angiogenesis. They may also have different effects depending on the timing of their production and release [17].

C-X-C motif chemokine ligand 8 (CXCL8), also known as interleukin-8, and chemokine C-C motif ligand 2 (CCL2), also known as chemoattractant protein 1 (MCP-1), are two chemokines that play crucial roles in post-ischemic stroke inflammation.

Furthermore, studies on murine models of ischemic stroke have shown that suppression of the CXCL8 gene reduces inflammation and promotes neurological recovery, suggesting therapeutic potential to modulate this pathway and improve outcomes in human patients, with a possible translational approach [18,19,20].

Instead, CCL2 plays a pivotal role in recruiting monocytes through binding to its C-C chemokine receptor type 2 (CCR2). This axis is crucial in transmitting neuroinflammatory signals between central nervous system cells. Indeed, the inhibition of CCL2/CCR2 has been demonstrated to attenuate inflammation, reduce blood–brain barrier permeability, and prevent the formation of cerebral oedema, consequently enhancing neurological outcomes. Moreover, elevated levels of CCL2 are associated with an increased risk of ischemic stroke and the exacerbation of ischaemic stroke progression [21,22,23,24].

Neuroinflammation is a complex mechanism, and in addition to the cells and mediators involved, it occurs through the brain structures that, when damaged, contribute to the spread of inflammatory injury: the dysfunction of the blood–brain barrier (BBB), and the simultaneous infiltration of peripheral immune cells (neutrophils and monocytes), is the condition that marks the spread of the inflammatory response, which becomes systemic, bringing an end to the isolation of the CNS.

The increase in the permeability of the BBB allows immune cells to reach the brain, thus exacerbating neurogenic inflammation and neuronal injury: it occurs through the deterioration of tight junction proteins, including occludin and claudin 5, induced by the phosphorylation and activation of metalloproteinases (MMP, particularly MMP-2 and MMP-9) or the translocation of caveolin-1 [25,26,27]. The production of reactive oxygen species (ROS) during ischaemia/reperfusion also contributes to BBB dysfunction. ROS damage endothelial cells and tight junction proteins and activate microglia, which in turn produce pro-inflammatory cytokines such as IL-1, IL-6 and TNF-α [26,28].

Once the blood–brain barrier has been damaged, neutrophils and monocytes migrate from the bloodstream to the brain: recent studies have demonstrated that, through the expression of damage-associated molecules (DAMPs) and matrix metalloproteinases (MMPs), neutrophils contribute significantly to endothelial cell death and further dysfunction of the BBB [29]. Moreover, infiltration of monocytes and neutrophils was observed in experimental stroke models, with a significant increase in these cells in the hemisphere affected by ischaemia [30]. In an ischaemic stroke, neutrophils are among the first leukocytes to be recruited in the affected cerebral region. These cells cross the blood–brain barrier and initially accumulate in the leptomeninges and perivascular spaces, subsequently infiltrating the brain parenchyma, particularly in areas of infarction [31,32].

In contrast, monocytes are recruited at a later stage than neutrophils, infiltrating the injury site within 24 h and reaching a maximum between 3 and 7 days post-stroke. Once in the ischaemic tissue, monocytes differentiate into macrophages. The function of these macrophages can be either pro-inflammatory or anti-inflammatory, depending on the microenvironment and the time elapsed since the ischemic event [33,34,35].

However, recent studies have demonstrated that, in addition to their well-established role in tissue damage through the production of reactive oxygen species and pro-inflammatory cytokines, neutrophils and monocytes may also benefit tissue repair and the resolution of inflammation: neutrophils can adopt different phenotypes (N1 and N2) with opposing functions; at the same time, monocytes can differentiate into macrophages with alternative activation characteristics, thereby promoting tissue repair [36,37].

Although ischemic stroke is a major cause of acute neuronal injury, it is not the only pathological condition that could trigger neuronal inflammation. A wide spectrum of central nervous system (CNS) disorders—including autoimmune diseases and infections—can converge on common endpoints characterized by neuronal damage, glial activation, and immune cell infiltration: inflammatory diseases such as multiple sclerosis (MS), and CNS infections all result in neuronal damage, but their pathophysiological mechanisms and immune responses differ fundamentally.

Multiple sclerosis is driven by chronic, autoimmune-mediated inflammation: here, microglia and infiltrating lymphocytes sustain a persistent inflammatory environment, with repeated cycles of demyelination and axonal degeneration. Microglia in MS also display dual roles: pro-inflammatory cytokines (i.e., IFN-γ, TNF-α) drive neurotoxicity, while anti-inflammatory mediators (i.e., IL-4, IL-13) support repair. Chronic microglial activation, mitochondrial dysfunction, and impaired remyelination underlie progressive neurodegeneration. Disease-modifying therapies in MS focus on dampening immune activation and promoting remyelination, with lessons increasingly informing ischemic stroke and infection management [38].

CNS infections (bacterial, viral etc.) elicit microglial activation via pathogen-associated molecular patterns. Microglia and astrocytes mount an acute inflammatory response, producing cytokines and chemokines that recruit leukocytes and disrupt the blood–brain barrier. While this response is essential for pathogen clearance, excessive or prolonged activation leads to neuronal injury and long-term consequences. Therapeutic approaches aim to balance anti-infective efficacy with neuroprotection, including modulation of microglial phenotype and cytokine production [39].

In summary, while neurodegeneration is a common endpoint, the initiating events, immune cell dynamics, and cytokine profiles differ across ischemic, inflammatory, and infectious CNS diseases. Although some insight into the management of neuroinflammation can be borrowed from autoimmune or infectious diseases, stroke therapy aimed at controlling inflammatory neuronal damage is still in its inception, and much progress remains to be made before it becomes a viable clinical option.

## 3. Inflammatory Circulating and Tissue Markers

In recent years, researchers have sought to identify circulating and tissue markers that can serve as valuable diagnostic and prognostic indicators. Great attention has been focused on the C-reactive protein (CRP). Elevated serum levels of high-sensitivity CRP (hs-CRP) at admission have a high predictive value for neurological deterioration, adverse functional outcome (modified Rankin scale ≥ 3 at 3 months), and an increased risk of death. Furthermore, elevated levels of CRP measured within 12 h of the onset of symptoms have been associated with an increased risk of adverse outcomes at 3 months [40,41].

A recent meta-analysis has demonstrated that elevated serum levels of hs-CRP at the time of admission are significantly associated with mortality, the risk of recurrence, and an adverse prognosis. Specifically, patients with higher levels of hs-CRP exhibit an increased risk of mortality (relative risk [RR] = 3.84), recurrence of stroke (RR = 1.88), and poor prognosis (RR = 1.77) [42]. Moreover, elevated levels of CRP at admission are indicative of poorer clinical outcomes and 90-day mortality in patients treated with mechanical thrombectomy. This finding suggests that CRP can be used as a biomarker to stratify risk in patients with acute ischemic stroke [43]. The diagnostic and prognostic importance of hsCRP is not insignificant, because it is an easily dosable marker available in all hospitals. When used appropriately, it not only provides information on the severity of ischemic damage at the time of diagnosis (regarding the onset and spread of neuroinflammatory injury) but also allows for the identification of a prognostic trajectory and the ability to predict worse post-ischemic recovery outcomes.

The High Mobility Group Box 1 (HMGB1) has emerged as an encouraging marker in recent studies on ischaemic stroke: it is a nuclear protein released by necrotic cells and actively secreted by platelets, immune, glial, and endothelial cells in response to ischemic damage and has an important role in sterile inflammation after an ischemic stroke. It binds to receptors such as the receptor for advanced glycation end products (RAGE) and toll-like receptors (TLRs), particularly TLR2 and TLR4. This triggers an inflammatory pathway involving the activation of microglial cells and the production of pro-inflammatory cytokines [44]. This “tissue” marker, after reaching the bloodstream, activates the complement system, which in turn promotes sterile inflammation. Moreover, the protein plays a pivotal role in regulating the permeability of the blood–brain barrier and modulating the peripheral immune response, contributing to the severity and outcome of stroke [45,46].

Furthermore, HMGB1 has been identified as a key mediator of microglial piroptosis and ferroptosis, which contribute to post-stroke neuronal damage: inhibition of HMGB1, through the use of glycyrrhizic acid, could exert neuroprotective effects [47].

HMGB1, through the Toll-Like Receptor 2—extracellular signal-regulated kinases 1/2—focal adhesion kinase (TLR2-ERK1/2-FAK) axis, promotes the migration of oligodendrocyte precursor cells (OPCs), thereby facilitating remyelination and the recovery of damaged brain tissue [48]. It also regulates the macrophage/microglial polarization towards phenotype M2, which is associated with tissue repair and reduction of ischaemic damage: when aptoglobin binds to HMGB1, the complexes resulting are quickly cleared by macrophage and microglial scavenger receptors; this process seizes HMGB1 and leads to an increase in the expression of M2-type markers, such as CD206, and a concomitant decrease in M1-type markers, including CD16 and CD32. This shift towards the M2 phenotype is crucial for reducing inflammation and promoting tissue repair [49].

Neopterin and myeloperoxidase are other potential markers of interest. Neopterin has been identified as an inflammatory biomarker produced by macrophages when they are activated in response to interferon-gamma (IFN-γ); it plays a crucial role in the pathophysiology of ischemic stroke, influencing inflammation and brain damage. Serum neopterin levels, in fact, are elevated in patients with ischemic stroke compared to healthy individuals, and these levels are associated with the severity of the disease and the volume of the infarction [50]. Neopterin has also been identified as a potential marker for adverse functional outcomes and mortality six months following an ischemic event, and is positively correlated with other inflammatory markers, including high-sensitivity C-reactive protein (hsCRP) [51].

Myeloperoxidase (MPO) is a pro-inflammatory enzyme primarily secreted by activated neutrophils and macrophages/microglia, playing a pivotal role in the inflammation and oxidative damage associated with ischemic stroke. MPO catalyzes the production of hypochlorous acid (HOCl) from hydrogen peroxide (H_2_O_2_) and chloride. This process is a significant contributor to the occurrence of oxidative stress and the oxidative modification of proteins. The production of HOCl by MPO can oxidize the HMGB1 protein, facilitating its secretion and release, which in turn activates neurogenic inflammation and exacerbates brain damage as well as blood–brain barrier dysfunction (BBB) [52]. Li et al. demonstrated that the ratio of MPO to high-density lipoprotein (HDL) (MPO/HDL) is a potential predictor of severity and functional outcome in patients with acute ischemic stroke. High MPO/HDL levels were associated with higher National Institutes of Health Stroke Scale (NIHSS) scores and worse 90-day functional outcomes [53]. There is a positive correlation between plasma concentrations of MPO and the severity of stroke, and unfavourable outcomes at three months. Furthermore, MPO has been associated with non-lacunar stroke subtypes and increased severity of stroke [54,55].

Over time, the role of MPO has become increasingly well defined, as it has been recognized that it plays a key role in the pathogenesis of ischaemic stroke: Wang et al. used Mendelian randomization to assess the causal effect of plasma MPO concentrations on the risk of ischaemic stroke, and the results showed that higher genetically determined levels of MPO were associated with an increased risk of ischemic stroke, particularly cardioembolic stroke (CEI) [56].

Indeed, a novel fluorescent probe activated by hypochlorous acid has been developed for the purpose of monitoring MPO activity in the ischemic brain in real time. This tool enables the visualisation of the endogenous activity of MPO and the identification of the compound AZD5904 as an effective inhibitor of MPO (it has reduced ischemic brain damage in mouse models) [57].

Therefore, the inhibition of MPO activity reduces inflammation and increases cellular protection in cases of ischemic stroke, thus improving neurological outcomes [58]. In summary, MPO contributes to post-stroke ischemic brain damage by producing HOCl, which induces oxidative stress and inflammation, thereby exacerbating neuronal damage and dysfunction of the BBB. Inhibition of MPO emerges as a promising therapeutic strategy to mitigate these deleterious effects.

The most recent research findings also indicate the existence of additional novel biomarkers associated with ischemic stroke, including microRNA (miRNA), extracellular vesicles (EV), and other inflammatory proteins. Several circulating miRNAs have significant diagnostic potential. Elevated levels of miR-19a-3p, let-7f, and miR-451a have been measured in patients with acute ischemic stroke, with areas under the curve (AUC) indicating adequate diagnostic ability. In addition, miR-106b-5p, miR-124, miR-155 and long non-coding RNAs (lncRNAs) H19 have been identified as promising biomarkers for the early diagnosis of ischemic stroke.

Extracellular vesicles (EV), particularly those originating from platelets and leukocytes, have been implicated in the enhancement of platelet reactivity and inflammation in patients with ischemic stroke. The combination of miR-19a-3p, platelet EV and leukocyte EV demonstrated a high diagnostic performance, with an area under the curve (AUC) of 0.893 [59,60,61].

In the field of precision medicine and “-omics sciences”, scientific research is making exciting progress: proteomic analysis identified serum inflammatory proteins such as tumor necrosis factor superfamily member (14TNFSF14 LIGHT), oncostatin M (OSM), NAD-dependent deacetylase sirtuin 2 (SIRT2), STAM-Binding Protein (STAMBP) and eukaryotic translation initiation factor 4E-binding protein 1 (4E-BP1), which are associated with adverse functional outcomes after stroke. These proteins are involved in regulating the NLR family pyrin domain-containing 3 (NLRP3) inflammasome and in signalling pathways including TNF, JAK/STAT, MAPK, and NF-κB [62]. Modern medicine is responding, with ever-increasing advances and progress, to the long-standing need to find diagnostic and prognostic markers for ischemic stroke: from already known markers that are gaining new attention (hsCRP), to newer protein whose pathogenic and neuroinflammatory role is becoming increasingly clear (HMGB1), to the products of immuno-inflammatory cells that mediate communication between organs, to transcription factors and inflammasome regulators; the complexity of this disease, which has always been known, is becoming increasingly characterized, highlighting the complex world of cellular crosstalk and the involvement of cellular systems as a whole, confirming, if ever there was any need, that ischemic stroke is a multisystemic disease and not only a cerebral vascular disease.

In clinical practice, the measurement of these markers is not yet systematic, and there is no uniform standardisation for their use in diagnosis or prognostic stratification, with the exception of high-sensitivity C-reactive protein. Therefore, as scientific knowledge and research in the field of neuroinflammation advances, we must try to gain as much information as possible to make their use feasible and useful in clinical settings.

## 4. Inflammatory Markers in Relation to Stroke Subtype and Its Severity

Several biomarkers appear to be associated with specific subtypes of ischemic stroke. These associations not only aid in understanding the underlying pathophysiology of the vascular accident but also provide potential targets for diagnosis and treatment. In stroke subtypes characterised by more extensive ischemic injury (cardioembolic stroke and large artery atherosclerosis), circulating levels of stromal cell-derived factor 1 alpha (SDF-1α) are generally higher compared to those observed in small vessel occlusion [63].

Further supporting the link between biomarkers and stroke recurrence, NT-pro-BNP is recognised as a marker of atrial and cardiac dysfunction, particularly in the context of atrial fibrillation and elevated cardioembolic risk. High serum concentration of NT-pro-BNP reflects underlying atrial cardiopathy or occult atrial fibrillation, both of which are established risk factors for recurrent cardioembolic events. The detection of NT-pro-BNP >1000 pg/mL has been correlated with a higher risk of recurrent cardioembolic stroke [64,65]. Concerning LAAS subtype, recent research has highlighted the importance of inflammation-related biomarkers such as interleukin-6 (IL-6) and macrophage migration inhibitory factor (MIF): IL-6 levels are significantly elevated in patients with LAAS compared to healthy individuals, and several meta-analyses confirm that the association between IL-6 and the risk of recurrent stroke is particularly pronounced in LAAS stroke compared to other subtypes, such as CEI. Elevated IL-6 levels predict the risk of recurrence and functional disability (mRS ≥ 2), even after adjustment for vascular risk factors and treatments, and has also been associated with faster stroke progression and poor functional outcomes within three months, in addition to being a predictor of carotid plaque severity, vulnerability, and progression, underscoring its central role in atherosclerosis [66,67,68]. Additionally, elevated inflammatory markers, including high-sensitivity C-reactive protein (hs-CRP), interleukin-6 (IL-6), and interleukin-1 receptor antagonist (IL-1Ra), have been consistently associated with poorer outcomes at three months post-stroke [69]. Migratory inhibitory factor (MIF) levels are significantly higher in LAAS compared to lacunar stroke. Elevated MIF is further linked to worse outcomes at three months and the development of malignant cerebral edema in patients with extensive hemispheric infarction [70,71].

A recent research project by our group has highlighted, as previously postulated, that each TOAST subtype of ischemic stroke has its own immuno-inflammatory profile, and that the different expression of inflammatory cytokines marks its peculiarity and partly explains its different pathogenesis: we reported that the cardioembolic subtype is characterized by a higher inflammatory burden expressed by significantly increased levels of TNF-α, IL-1 and IL-6, and reduced IL-10 levels compared to both the LAAS and lacunar subtypes [72]: this finding, in agreement with other studies, supports further the hypothesis that inflammation plays a pivotal role in the pathogenesis of ischemic stroke, but also confirms the presence of a peculiar immunoinflammatory pattern for each stroke subtype. The marked inflammatory background that distinguishes the cardioembolic infarct from other subtypes agrees with previous reports indicating the higher clinical and prognostic severity of this subtype [73]. This could result from an increased recall of inflammatory cells, such as polymorphonuclear cells, as suggested in the study by Licata et al. [74].

The presence of an essential inflammatory component in the CEI subtype is partly due to the close association between this subtype and atrial fibrillation. The latter represents the leading cause of cardioembolic stroke: several pieces of evidence support the role of inflammation in the pathogenesis and maintenance of atrial fibrillation [75,76]. The role of inflammation in atrial fibrillation derives from the observation of its high incidence in patients with overt inflammatory diseases of cardiac origin (myocarditis, pericarditis) and non-cardiac (pneumonia and inflammatory bowel diseases). Regardless of whether atrial fibrillation is the cause or consequence of an inflammatory mechanism, it is related to oxidative stress associated with myocardial infiltration by inflammatory cells, and consequent release of ROS. The inflammatory environment leads to the activation of RAAS (renin–angiotensin–aldosterone system) and, thus, to the activation of NADPH (nicotinamide adenine dinucleotide phosphate hydrogen) oxidase. Consequently, this cascade triggers the activation of the TGF-β (transforming growth factor-beta) pathway and the structural and electrical remodelling of the myocardium. Thus, an increased expression of various inflammatory cytokines and chemokines, such as IL-1, IL-6, and TNF-α, occurs. Therefore, the close association between atrial fibrillation and cardioembolic stroke could explain the immune-inflammatory phenotype that characterises CEI.

TNF-α contributes to the instability of atherosclerotic plaques and thrombosis, which are central mechanisms in atherothrombotic stroke. Persistently high levels of tumour necrosis factor-alpha (TNF-α >24 pg/mL) are associated with an increased incidence of atherothrombotic stroke, highlighting the significance of systemic inflammation. In an extensive real-world cohort study of over 5000 stroke patients, persistently elevated TNF-α levels above 24 pg/mL after stroke recurrence were independently associated with atherothrombotic stroke subtype, with an adjusted odds ratio of 21.26 (95% CI: 12.42–37.59) for recurrence, indicating a strong relationship between chronic inflammation and atherothrombotic events [64].

Metabolomic research has identified additional biomarkers to distinguish LAAS from cardioembolic infarction (CEI). Targeted analysis using liquid chromatography-tandem mass spectrometry on serum samples from 346 participants revealed elevated concentrations of several amino acids and related compounds in LAAS compared to CEI. These included lysine, serine, threonine, kynurenine, putrescine, and lysophosphatidylcholine acyl C16:0 (LPC C16:0) [77]. In this regard, advances in metabolomics have enabled the differentiation of subtypes of ischemic stroke by identifying specific target molecules. Li et al. identified 12,13-dihydroxy-9Z-octadecenoic acid (12,13-diHOME) as a pivotal metabolite that differentiates extensive artery atherosclerosis (LAAS) from small vessel disease (SVD). Employing high-resolution mass spectrometry to analyse plasma metabolic profiles of ischemic stroke patients classified according to the TOAST criteria, the study revealed 26 differential metabolites, with 12,13-diHOME emerging as the most significant. This metabolite demonstrated an area under the curve (AUC) of 0.822 and an accuracy of 77.8% in predicting stroke subtype [78]. Strokes classified under ‘Other Determined Etiology’ (ODE) within the TOAST framework encompass a variety of less common causes, each associated with specific biomarkers. For example, elevated levels of matrix metalloproteinase-9 (MMP-9) are linked to arterial dissection, contributing to extracellular matrix degradation and aneurysm formation. This marker is handy in differentiating dissection from other causes, especially in younger patients.

High D-dimer concentrations are strongly associated with cancer-related coagulopathy, while increased P-selectin levels indicate platelet dysfunction and coagulation disorders, playing a key role in thrombus formation. Specifically, elevated levels of MMP-9 and D-dimer have been associated with greater neurological severity (higher NIHSS scores) and poorer functional outcomes (mRS ≥ 3 at 3 months) in ODE subtypes, such as arterial dissection and cancer-associated coagulopathy. Additionally, altered serotonin levels have been observed in strokes related to migraine and drug use, suggesting serotonin’s role in vasoconstriction and endothelial dysfunction [79]. Beyond classification, several studies have explored the relationship between inflammatory biomarkers and stroke severity, as measured by the National Institutes of Health Stroke Scale (NIHSS) and the modified Rankin Scale (mRS). Systemic inflammatory markers such as the neutrophil-lymphocyte ratio (NLR), platelet-lymphocyte ratio (PLR), systemic immune inflammation index (SII), and segmented neutrophil-monocyte ratio (SeMo) have been consistently associated with the severity of acute ischemic stroke. Notably, elevated levels of these markers correlate with moderate to severe strokes (NIHSS ≥ 6) and with severe disability or death within 30 days (mRS ≥ 4). Numerous trials have demonstrated that elevated baseline values of NLR, PLR, and SII are independent predictors of NIHSS ≥ 6 and mRS ≥ 4, particularly in patients with LAAS and SVD, compared to other TOAST subtypes. Specifically, NLR demonstrates the best predictive performance for unfavourable post-thrombolysis outcomes in the LAAS (AUC 0.702) and SVD (AUC 0.750) subtypes, with significant adjusted odds ratios for poor functional outcome at 90 days. SII and PLR are also significantly associated with greater severity and worse prognosis, though their predictive performance is slightly lower than that of NLR. 

In cardioembolic subtypes, this association is present but less pronounced compared to LAAS and SVD. Regarding the ODE subtype (Other Determined Aetiology), the available evidence remains limited. It does not demonstrate a comparably strong association between these biomarkers and stroke severity, unlike the atherothrombotic and lacunar subtypes [80,81,82,83].

Huang et al. also confirmed that high SII is significantly associated with adverse clinical outcomes, including worse functional outcome (mRS 3), mortality and haemorrhagic transformation [84]. Increased serum levels of neuronal injury biomarkers such as β-synuclein (β-syn), neurofilament light chain (NfL), and glial fibrillary acidic protein (GFAP) have shown significant correlations with higher NIHSS scores during hospitalisation and elevated modified Rankin Scale (mRS) scores at follow-up, reflecting a worse functional prognosis. Indeed, patients exhibiting elevated concentrations of these biomarkers demonstrated reduced clinical improvement within the first 24 h and experienced poorer long-term recovery [85]. Furthermore, other markers indicative of neuronal damage—namely myelin basic protein (MBP), neuron-specific enolase (NSE), and S100β protein—have been linked to greater stroke severity at baseline and larger lesion volumes on 24 h CT scans. Interestingly, early dynamic changes in MBP and S100β levels within the initial 24 h following stroke onset have been associated with more favourable long-term functional outcomes [86].

## 5. Clinical Implications: Diagnosis, Prognosis and Treatment

The integration of inflammatory and neuronal injury biomarkers into the management of ischemic stroke is redefining diagnostic, prognostic, and therapeutic strategies, with significant clinical implications. Several studies have shown that multimarker panels—including high-sensitivity C-reactive protein (hs-CRP), interleukin-6 (IL-6), neutrophil-to-lymphocyte ratio (NLR), MMP-9, GFAP, NfL, and β-syn —enhance the prediction of disability, mortality, and risk of recurrence compared to traditional clinical factors alone [68,87,88,89].

The association between elevated levels of hs-CRP and IL-6 and the risk of recurrence is particularly pronounced in atherothrombotic and lacunar stroke subtypes, even among patients adhering to guideline-recommended secondary prevention strategies [90]. The clinical impact of biomarkers extends to the possibility of personalising secondary prevention: the combination of serum biomarkers and advanced imaging parameters (e.g., vessel wall imaging, plaque vulnerability assessment) enables more accurate risk stratification. It can guide the intensification of therapeutic strategies in high-risk patients [91,92,93].

As said above, neuroinflammation represents a crucial pathogenic hub in ischemic stroke, influencing the progression of neuronal damage, blood–brain barrier dysfunction, and the reparative response. Activation of microglia, astrocytes, and infiltration of peripheral immune cells promote the production of pro-inflammatory cytokines (TNF-α, IL-1β, IL-6), MMPs, and other molecules that amplify tissue damage and vascular dysfunction. However, neuroinflammation also plays a protective role during the recovery phase, suggesting that therapeutic intervention should be targeted and appropriately timed. From a therapeutic perspective, modulation of neuroinflammation is an active area of research. Some chemical agents, such as edaravone and dexborneol, have demonstrated clinical efficacy in Phase III studies in China, acting on both oxidative stress and the inflammatory response. Other experimental approaches include the inhibition of specific cytokine pathways (e.g., JAK/STAT3, NF-κB) and the use of immunomodulatory agents. However, further randomized trials are needed to define their clinical role, since the clinical implications associated with inhibiting such complex inflammatory platforms could have negative consequences in terms of defence against infections and cancers.

The prognostic importance of inflammatory and neuronal biomarkers is now well established: high levels of IL-6, TNF-α, MMP-9, GFAP, NfL, and β-synuclein are associated with worse functional outcomes, a higher risk of relapse, and increased mortality. They can guide the selection of patients for more aggressive therapeutic strategies or enrollment in clinical trials [94,95,96]. Concurrent studies are underway to investigate other agents, including MCC950, which inhibits the NLRP3 inflammasome, and minocycline, which reduces microglial activation. These studies are currently in the preclinical and clinical trial phases, with initial results encouraging [8]. Inflammatory and neuronal damage biomarkers represent promising tools for treatment personalization and patient selection for targeted anti-inflammatory therapies: cytokine inhibitors, particularly targeting IL-1β (such as anakinra and canakinumab), or TNF-α (such as etanercept), have been evaluated in preclinical and proof-of-concept clinical studies, demonstrating the ability to reduce the extent of ischemic damage and blood–brain barrier dysfunction. However, cytokine neutralisation may lead to immunological side effects, and patient selection, as well as therapeutic windows, remain a challenge [97,98].

Recent investigations have elucidated the pivotal role of S100A8/A9^hi^ neutrophils in mediating endothelial cell apoptosis and promoting lymphocyte infiltration following ischemic stroke, as revealed by single-cell RNA sequencing (scRNA-seq). These neutrophils contribute substantially to blood–brain barrier (BBB) disruption, thereby exacerbating neuroinflammation. Therapeutic targeting of S100A8/A9 with inhibitors, such as paquinimod, has demonstrated neuroprotective effects, including the attenuation of neuronal damage and reduced lymphocyte infiltration, positioning this approach as a promising strategy to mitigate post-stroke neurological deficits [29]. In parallel, emerging evidence highlights the complement cascade, particularly the component C3, as a critical mediator of post-ischemic neuroinflammation. In vivo studies employing targeted C3 inhibition via CR2-fH have reported significant improvements in both radiological and functional outcomes. Collectively, these findings highlight the therapeutic potential of modulating distinct inflammatory pathways—namely, neutrophil activity and complement activation—to mitigate neuroinflammation and promote recovery following ischemic stroke [99].

Moreover, extensive research has been undertaken to evaluate corticosteroids as a therapeutic approach in ischemic stroke, explicitly focusing on their role in modulating neuroinflammation. Preclinical studies have suggested potential benefits, including stabilisation of the blood–brain barrier, reduction in vasogenic oedema, and modulation of the immune response; however, these effects have not been consistently confirmed in clinical studies in humans. Recent evidence, including the randomised MARVEL trial, has evaluated the use of low-dose methylprednisolone (2 mg/kg/day) as an adjunct to endovascular thrombectomy in patients with large-vessel occlusion stroke. The results did not demonstrate a significant improvement in disability at 90 days. However, they showed a reduction in mortality and symptomatic intracranial haemorrhages, as well as a lower incidence of pneumonia and circulatory failure. However, the effect on primary functional outcomes remains neutral, and the generalizability of the findings is limited by the exclusively Chinese patient population and the need for further validation in other settings. A recent meta-analysis confirmed that corticosteroids may reduce mortality at 3 months following ischemic stroke; however, this benefit does not extend to functional outcomes and is accompanied by an increased risk of infectious complications. Consequently, the American Heart Association/American Stroke Association guidelines recommend against the routine use of high-dose corticosteroids in ischemic stroke due to the lack of demonstrated benefit and the potential for damage [100,101].

This therapeutic approach has, in fact, already been explored in the field of cardiovascular risk linked to immunosenescence: in a review by our group [102], we examined all trials that had evaluated the administration of anti-inflammatory drugs (colchicine, canakinumab, methotrexate, statins) in patients with coronary artery disease, and we found that the modulation of sterile inflammation is a real chimera in 21st-century medicine, because while it aims to reduce the harmful effects of chronic inflammation, it also exposes patients to significant risks related to the development of infections, cancers and autoimmune diseases. Therefore, by applying this approach to ischemic stroke, researchers are working to modulate inflammatory processes in an increasingly tailored manner; however, the road ahead is still long and fraught with pitfalls.

## 6. New Insights and Future Directions

Emerging evidence is reshaping our understanding of neuroinflammation in ischemic stroke, underscoring its multifaceted role in both injury and repair. Recent scientific literature also identifies the nuclear factor erythroid 2–related 2-related factor 2 (Nrf2) signalling pathway as a promising therapeutic target. Nrf2 activation in glial cells suppresses oxidative stress and downstream inflammatory cascades, including the NF-κB and NLRP3 inflammasome pathways, suggesting that Nrf2 activators may offer neuroprotection by dampening neuroinflammation [103].

Kuo et al. have elucidated the pivotal role of the Nrf2/heme oxygenase-1 (HO-1) signalling axis in regulating microglial polarisation and mitigating ischemic brain injury. Under ischemic conditions, Nrf2 activation within microglia induces HO-1 expression, thereby promoting a phenotypic shift from a pro-inflammatory state, marked by elevated levels of CD68 and IL-1β, to an anti-inflammatory phenotype characterised by increased CD206 expression. This polarization shift significantly attenuates neuroinflammatory responses and constrains secondary neuronal damage. Evidence from microglia-specific Nrf2 knockdown models reveals exacerbation of ischemic injury and an expansion of pro-inflammatory microglial populations, underscoring the indispensable role of Nrf2 in modulating neuroinflammation. Conversely, pharmacological activation of the Nrf2/HO-1 pathway confers neuroprotection, even in the presence of comorbidities such as diabetes, highlighting its therapeutic potential [104].

Beyond its immunomodulatory functions, Nrf2 serves as a key regulator of ferroptosis, a form of regulated cell death driven by iron-dependent lipid peroxidation, which is notably prominent following cerebral ischemia. Nrf2 activation enhances the expression of xCT (SLC7A11), a cystine/glutamate antiporter, and glutathione peroxidase 4 (GPX4), a selenoenzyme that catalyzes the reduction in lipid hydroperoxides. This upregulation enhances intracellular glutathione synthesis and strengthens antioxidant defences, thereby inhibiting ferroptotic cell death and preserving neuronal viability. Both in vitro and in vivo studies have demonstrated that Nrf2 agonists, such as oltipraz, upregulate xCT and GPX4, thereby improving iron overload and reducing neuronal injury. In contrast, genetic ablation or pharmacological inhibition of Nrf2 exacerbates ferroptosis and worsens ischemic outcomes. Collectively, these findings highlight the central role of Nrf2 in orchestrating anti-inflammatory and anti-ferroptotic responses, and support the therapeutic potential of Nrf2 activators —including sulforaphane, oltipraz, and various phytochemicals—in attenuating oxidative stress, limiting neuroinflammation, and improving neurological outcomes in ischemic stroke models [105,106,107].

Another field offering interesting prospects and exciting avenues of research about cerebrovascular health is the gut microbiome, which has emerged as a crucial modulator of host physiology, extending its influence well beyond local gastrointestinal functions to systemic processes, including immune regulation and neural health. Increasing evidence suggests that the gut microbiota plays a central role in shaping neuroinflammatory responses, thereby influencing the pathophysiology and outcome of ischemic stroke. The bidirectional communication between the gut and the brain, commonly referred to as the gut–brain axis, provides a mechanistic framework linking dysbiosis to post-stroke inflammation, blood–brain barrier integrity, and neuroimmune signalling. The gut microbiome and metabolic dysregulation are now recognized as modulators of neuroinflammatory responses, with lipopolysaccharide-driven inflammation linking systemic and cerebral injury. Nuszkiewicz et al. highlight that the role of dysregulated adipokines in obesity amplifies inflammatory responses, thereby increasing both the risk and severity of acute ischemic stroke. Disruptions in the gut microbiota, particularly those leading to increased gut permeability and the translocation of bacterial products such as lipopolysaccharides (LPS), further drive neuroinflammation and worsen stroke outcomes. The LPS-induced neuroinflammatory cascade serves as a bridge between systemic metabolic diseases and central nervous system injury in stroke [108]. Microbiota-derived metabolites, especially short-chain fatty acids (SCFAs), can modulate neuroinflammation: SCFAs are protective, while elevated trimethylamine N-oxide (TMAO) is associated with increased thrombotic risk and worse outcomes [109,110].

The assessment of inflammatory and neuronal biomarkers holds significant diagnostic and prognostic value in ischemic stroke, marking a progressive and steady advance towards precision medicine approaches aimed at minimizing disability and reducing the risk of recurrence. Incorporating these biomarkers into routine clinical practice promises to enhance patient stratification and guide personalized therapeutic interventions. However, the widespread implementation of such tools needs rigorous prospective validation and the establishment of standardized methodologies to ensure consistency and reliability across diverse clinical settings. In addition, given the central role of neuroinflammation in the pathophysiology of ischemic stroke, there is a pressing need for continued research focused on elucidating the complex inflammatory mechanisms underlying stroke progression and recovery. This growing body of evidence not only deepens our understanding of stroke pathophysiology but also opens new perspectives for the development of innovative therapeutic strategies that specifically target neuroinflammatory pathways. By prioritizing neuroinflammation as a therapeutic target, future studies have the potential to significantly improve clinical outcomes and transform the management of ischemic stroke, ultimately moving toward more effective, tailored treatments for affected patients.

To summarise, beyond the scientific interest that these individual biomarkers have, it remains to be seen how they can be applied in clinical practice and reproduced in everyday situations, bearing in mind that this field is still subject to many limitations and that, unfortunately, there is still no biomarker for ischaemic stroke that can replace radiodiagnostics or clinical examination in the diagnosis (especially early diagnosis) of the disease. For example, glial fibrillary acidic protein (GFAP) has demonstrated a sensitivity ranging from 73% to 80% and a specificity between 77% and 97% for distinguishing ischaemic from haemorrhagic stroke within the first 3 to 24 h after symptom onset. A multimarker panel comprising S100β, matrix metalloproteinase-9 (MMP-9), D-dimer, brain-derived neurotrophic factor (BDNF) and von Willebrand factor (vWF) showed a sensitivity of 92% and a specificity of 93% in diagnosing ischaemic stroke within six hours, although large-scale validation is still required.

MMP-2 showed a sensitivity of 62.5% and a specificity of 88.5%, and Thymic stromal lymphopoietin (TSLP) demonstrated a sensitivity of 66.7% and a specificity of 96.2%.

Although D-dimer and fibrinogen are easily measurable and widely available, their specificity is limited as elevated levels are also observed in other thrombotic conditions.

In conclusion, neither of the biomarkers currently available exhibits the required level of diagnostic accuracy in terms of both sensitivity and specificity to support an independent diagnosis of acute ischaemic stroke in routine clinical practice. Although multimarker panels have demonstrated superior performance, their clinical applicability is limited by methodological variability, a lack of standardisation and the need for robust external validation across diverse populations and clinical settings.

However, emerging evidence suggests that integrating biochemical biomarkers into comprehensive, multimodal, diagnostic strategies, potentially enhanced by advanced computational tools and machine learning algorithms, could significantly improve diagnostic accuracy and reduce the time taken to start treatment.

Future research should therefore focus on identifying, validating and implementing individual or combined biomarker signatures that can play a pivotal role in the early, accurate and tailored diagnosis of ischaemic stroke, thereby improving patient outcomes and enabling more precise therapeutic approaches.

## Figures and Tables

**Figure 1 ijms-26-09801-f001:**
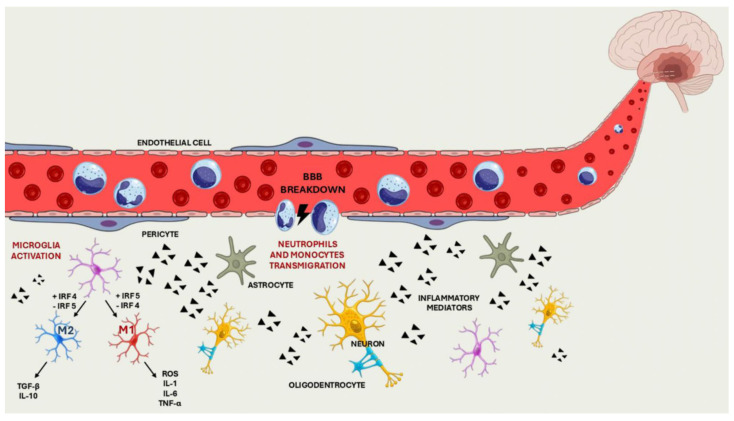
Breakdown of the BBB, migration of the immune cells into the CNS and activation of the cell differentiation pathway.

**Table 1 ijms-26-09801-t001:** TOAST classification.

Subtype	Pathophysiological Mechanism	Diagnostic Criteria	Etiology	Clinical/Imaging Features
**Large-Artery Atherosclerosis (LAAS)**	Artery-to-artery embolism or in situ thrombosis due to atherosclerosis	≥50% stenosis or occlusion of a large extracranial/intracranial artery	Atherosclerosis of carotid, vertebral, or cerebral arteries	Cortical or subcortical infarcts, often wedge-shaped; history of vascular risk factors; recurrent transient ischemic attacks (TIAs)
**Cardioembolism (CEI)**	Embolism from a cardiac source	High-risk cardiac source identified; absence of significant large-artery disease and lacunar pattern	Atrial fibrillation, recent MI, LV thrombus, valvular disease, cardiac tumours, PFO with DVT	Sudden onset; large cortical infarcts; hemorrhagic transformation common; multiple territory infarcts
**Small-Vessel Occlusion (SVO)**	Lipohyalinosis or microatheroma of small penetrating arteries	Classic lacunar syndrome; infarct <1.5 cm in deep brain structures; exclusion of CEI and LAAS	Hypertensive arteriopathy, diabetes-related small vessel disease	Pure motor or sensory stroke; internal capsule, thalamus, basal ganglia infarcts; absence of cortical signs
**Stroke of Other Determined Etiology (ODE)**	Uncommon or rare causes with known mechanisms	Specific cause identified through focused diagnostic testing	Dissection, vasculitis, prothrombotic conditions, genetic/metabolic disorders, infective endocarditis	Variable, depending on the aetiology; often seen in younger patients; requires an extensive diagnostic workup.
**Stroke of Undetermined Etiology** **(UDE)**	Unknown or competing mechanisms	No identified cause after complete diagnostic workup; >1 potential cause	Paroxysmal AF not detected, silent atherosclerosis, unrecognized hypercoagulable state	Heterogeneous presentation

## Data Availability

No new data has been created in this manuscript.

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
