# Peer review of "Tracing Inflammation in Ischemic Stroke: Biomarkers and Clinical Insight"

_ijms, 2025, doi:10.3390/ijms26199801_

Round 1
Reviewer 1 Report
Comments and Suggestions for Authors
Authors present a comprehensive and critical examination of the current literature evidence on the inflammatory response of ischemic stroke, focusing on molecular mechanisms and biomarkers of neuroinflammation, their importance in prognostic stratification, and their implications for novel therapeutic targets.
The review is lengthy and relies on well-known information. Authors should focus and clearly state the purpose of this type of review. The authors need to improve the reliability and novelty of the article. The methodology of the review should be shown. The type of review, principles of its compilation, keywords, and criteria for including articles for analysis are not specified.
There are errors in the methodological approach and interpretation of results. The authors examine biochemical markers of tissue damage that may indicate brain tissue necrosis and the development of inflammation. The authors present general information about the inflammatory response to tissue injury.
In the context of this review, it is necessary to consider the role of the blood-brain barrier, which limits both the recognition of damaged brain tissue and the spread of damage markers beyond the brain.
It is necessary to focus on the inflammation that is initiated by cerebral ischemia, to show the differences from other situations (for example, viral encephalitis, multiple sclerosis, etc.).
The authors need to better systematize the biomarkers. Tissue-specific biomarkers should be identified, sensitivity and specificity for ischemic stroke should be shown, if this can be determined. Inflammatory markers should also be identified and their diagnostic value should be shown.
A general conclusion should be made based on the literature analysis conducted.
There is no reference to Figure 1 in the text.
My general comment is that the manuscript in its current form is not ready for publication due to its low readability and scientific value.
Author Response
Comment 1: The review is lengthy and relies on well-known information. Authors should focus and clearly state the purpose of this type of review. The authors need to improve the reliability and novelty of the article. The methodology of the review should be shown. The type of review, principles of its compilation, keywords, and criteria for including articles for analysis are not specified.
Response: Thank you for this comment. The work is a narrative review; therefore, the authors selected the most recent scientific literature on the topic and analysed it to provide a summary for the reader, update, and inform them on the main new findings relating to neuroinflammation in ischemic stroke and the role of single inflammatory mediators. To do this, the authors used the PubMed and Google Scholar search platforms, entering the keywords specified immediately after the abstract into the query box and, for some of the markers whose diagnostic or prognostic role they explored in depth, conducting a specific search. To this end, and to make the type of review clear to readers from the beginning, I have added the word “narrative” before “review” at the beginning of the manuscript, highlighted in red.
Comment 2: There are errors in the methodological approach and interpretation of results. The authors examine biochemical markers of tissue damage that may indicate brain tissue necrosis and the development of inflammation. The authors present general information about the inflammatory response to tissue injury. In the context of this review, it is necessary to consider the role of the blood-brain barrier, which limits both the recognition of damaged brain tissue and the spread of damage markers beyond the brain. It is necessary to focus on the inflammation that is initiated by cerebral ischemia, to show the differences from other situations (for example, viral encephalitis, multiple sclerosis, etc.).
Response: Thank you for your stimulating suggestion. Indeed, having discussed damage to the BBB and the effects that this damage produces, we omitted to specify the difference between inflammation-based ischaemic damage and non-ischaemic inflammatory damage from a molecular point of view, so we have added a section to the manuscript to clarify this, highlighted in red in the main text.
Comment 3: The authors need to better systematize the biomarkers. Tissue-specific biomarkers should be identified, sensitivity and specificity for ischemic stroke should be shown, if this can be determined. Inflammatory markers should also be identified and their diagnostic value should be shown.
Response: Thank you for clarifying that. In paragraph 3 of the text, which deals with circulating and tissue markers, I have highlighted the origin of the markers in red, specifying whether they are tissue markers (“at the beginning”, because the determination always takes place after they have passed into the circulation) or serum markers. Sensitivity, specificity and other biostatistical information relating to the markers have been reported where available; where not reported, this means that they are not provided in the scientific literature.
Comment 4: A general conclusion should be made based on the literature analysis conducted.
Response: Thanks for the suggestion. We have added a textual part in the "Introduction" paragraph, although it is a narrative review.
Comment 5: There is no reference to Figure 1 in the text.
Response: Thanks for the suggestion. We added the reference to the Figure 1, highlighted in red in the main text.
Reviewer 2 Report
Comments and Suggestions for Authors
This review presents ischemic stroke as an inflammatory disease, highlighting how immune cells and markers contribute before and after vascular injury, and discusses emerging research on potential biomarkers that could enable earlier diagnosis and improve treatment outcomes. The comments on this manuscript are as follows:
- The topic is timely and clinically applicable and the review gives useful information on the role of inflammation in ischemic stroke
- The review only includes one figure. I would strongly recommend to authors to include more schematic figures or summary tables for better representation
- I suggest authors to include new biomarkers (exosomal, microRNA, multi-omics) and clearly state how well each is validated
- In terms of clinical translation, the review should distinguish between markers that are clinically applicable now and those that are experimental.
Author Response
Comment 1: This review presents ischemic stroke as an inflammatory disease, highlighting how immune cells and markers contribute before and after vascular injury, and discusses emerging research on potential biomarkers that could enable earlier diagnosis and improve treatment outcomes. The comments on this manuscript are as follows:
- The topic is timely and clinically applicable and the review gives useful information on the role of inflammation in ischemic stroke
- The review only includes one figure. I would strongly recommend to authors to include more schematic figures or summary tables for better representation
- I suggest authors to include new biomarkers (exosomal, microRNA, multi-omics) and clearly state how well each is validated
- In terms of clinical translation, the review should distinguish between markers that are clinically applicable now and those that are experimental.
Response:
2: thanks for the suggestion; we have included a table and a figure to make the content more schematic and easily accessible. Since the creative and production effort (the image is original and created entirely by us, the authors) was considerable, considering that the image contains many molecular details described in the text, we would kindly ask you to consider whether the number of images and tables is sufficient for this manuscript;
3: thanks for the comment; in the paragraph 3, we have included references to the -omics sciences and mentioned the latest findings in terms of neuroinflammation (highlighted in red in the main text);
4: thank you for your contribution. We have added a section at the end of paragraph 3 specifying which markers are available and routinely used in clinical practice and which are not yet available, with the hope that standardisation will be achieved as soon as possible.
Round 2
Reviewer 1 Report
Comments and Suggestions for Authors
The article has been improved, but remains lengthy. A conclusion needs to be reached regarding the feasibility of using biochemical markers for stroke diagnosis, and the most specific ones should be identified.
Author Response
Comment 1: The article has been improved, but remains lengthy. A conclusion needs to be reached regarding the feasibility of using biochemical markers for stroke diagnosis, and the most specific ones should be identified.
Response: Thank you for the additional insight. I have added a conclusion that summarises the actual applicability of the most studied markers, going beyond mere scientific suggestions, at least for the markers for which we have this data. I have also shortened the manuscript by removing some text, making it more readable and accessible.